# Targeting Estrogens and Various Estrogen-Related Receptors against Non-Small Cell Lung Cancers: A Perspective

**DOI:** 10.3390/cancers14010080

**Published:** 2021-12-24

**Authors:** Radhashree Maitra, Parth Malik, Tapan Kumar Mukherjee

**Affiliations:** 1Department of Biology, Yeshiva University, 500 W 185th Street, New York, NY 10032, USA; 2Montefiore Medical Center and Albert Einstein College of Medicine, 1300 Morris Park Avenue, Bronx, NY 10461, USA; 3School of Chemical Sciences, Central University of Gujarat, Gandhinagar 382030, Gujarat, India; parthmalik1986@gmail.com; 4Department of Biotechnology, Maharishi Markandeshwar University Mullana, Ambala 133207, Haryana, India

**Keywords:** pre/postmenopausal women, non-small cell lung cancers (NSCLCs), estrogens, estrogen receptors (ERs), G-protein-coupled ERs (GPERs), epidermal growth factor receptors (EGFRs), estrogen-related receptors (ERRs), anti-estrogen/ER/GPER/EGFR/ERR therapies against NSCLCs

## Abstract

**Simple Summary:**

Exogenous and endogenous estrogens and associated receptors modulate signaling pathways with biochemical events implicated in non-small cell lung cancer (NSCLC) manifestation. The diversity of biochemical interactions initiated by estrogens is rigorous, regulated via distinct estrogen-associated receptors. While estrogen receptor beta (ERβ) is overexpressed in 60–80% of NSCLCs irrespective of gender, the recognition of transmembrane G-protein-coupled estrogen receptor (GPER) creates several interfaces of estrogen-interception-driven aggressive NSCLC manifestation. There is still room for understanding the crux of ER–EGFR (epidermal growth factor receptor) interactions considering the recent clinical trials revealing a synergistic anti-NSCLC response. With such insights, this manuscript presents a comprehensive discussion on the sequential biochemical events in estrogen-activated cell signaling pathways in NSCLC complications with a focus on the ER/GPER/EGFR/ERR regulatory mechanism alongside the NSCLC treating anti-estrogen targeted therapies.

**Abstract:**

Non-small cell lung cancers (NSCLCs) account for ~85% of lung cancer cases worldwide. Mammalian lungs are exposed to both endogenous and exogenous estrogens. The expression of estrogen receptors (ERs) in lung cancer cells has evoked the necessity to evaluate the role of estrogens in the disease progression. Estrogens, specifically 17β-estradiol, promote maturation of several tissue types including lungs. Recent epidemiologic data indicate that women have a higher risk of lung adenocarcinoma, a type of NSCLC, when compared to men, independent of smoking status. Besides ERs, pulmonary tissues both in healthy physiology and in NSCLCs also express G-protein-coupled ERs (GPERs), epidermal growth factor receptor (EGFRs), estrogen-related receptors (ERRs) and orphan nuclear receptors. Premenopausal females between the ages of 15 and 50 years synthesize a large contingent of estrogens and are at a greater risk of developing NSCLCs. Estrogen—ER/GPER/EGFR/ERR—mediated activation of various cell signaling molecules regulates NSCLC cell proliferation, survival and apoptosis. This article sheds light on the most recent achievements in the elucidation of sequential biochemical events in estrogen-activated cell signaling pathways involved in NSCLC severity with insight into the mechanism of regulation by ERs/GPERs/EGFRs/ERRs. It further discusses the success of anti-estrogen therapies against NSCLCs.

## 1. Introduction

Estrogens are a group of steroid hormones having multiple physiological roles including the development of primary and secondary reproductive organs as well as that of nongonadal organs such as lungs [1,2,3]. In the mammalian body, gonads (ovary/testis) synthesize and release estrogens into the circulation and are thus considered as a primary source of estrogens [4]. A low level of estrogens is also generated locally at various tissue levels including that of lungs [5,6,7]. Under normal physiological conditions, estrogens are involved in lung development [8,9]. 17-β-estradiol (E2) is the major estrogen present in the circulating female blood with a half-life of approximately 3 h subsequent to which it undergoes a rapid and irreversible oxidation into the estrogen metabolites, estrone (E1) and estriol (E3), by specific enzymes present in the liver [10]. A number of other estrogen metabolites such as 2-OH-E2 and 4-OH-E2 are also endogenously produced in the human body [11]. Besides endogenous estrogens and their metabolites, the human body is exposed to exogenous estrogens such as synthetic estrogen ethinyl estradiol (EE), either as a constituent of contraceptive pills [12,13] or hormone response therapy (HRT) (Table 1) [14,15]. Apart from endogenous and exogenous estrogens, the human body is exposed to phytoestrogens such as genistein through plant-derived food materials [16]. Finally, a large number of environmental agents called xenoestrogens show potential estrogenic and endocrine disruptive properties [17]. Endocrine disruptors are molecules that regulate the expression of multiple genes, either via direct interaction with the ERE or indirectly by interacting with transcription factors, including members of activator protein-1 (AP-1), nuclear factor-κB (NF-κB), signal transducer and activator of transcription (STAT) and families of specificity protein (SP-1), or by modifying the estrogen metabolism [18,19,20]. Figure 1 summarizes the endogenous and exogenous sources of estrogens along with their implicit metabolic fates while Table 1 lists the various estrogen sources of each domain.

Collectively, exposure of the human body toa high level of natural or synthetic estrogens/phytoestrogens/xenoestrogens or different estrogen metabolites such as 4-OH-E2 has adverse physiological effects including the development of various cancers. A high serum level of estrogens is related to the troublesome prognosis of non-small cell lung cancers (NSCLCs) besides enhancing the severity [21]. The common links between estrogenic metabolism and tobacco combustion aggravate the carcinogenic actions in the lungs. The constituents of cigarette smoke activate cytochrome P450 1B1 (CYP1B1), the enzyme mitigated in estrogen metabolism along with the synthesis of corresponding catecholic derivatives. Intermediates and products formed therein accumulate as reactive oxygen species (ROS), concurrently prevailing as a DNA adduct that collectively tampers the genetic material. Interactions of estrogens with cigarette smoke constituents are driven via ER interception, which forms genotoxic metabolites including 4-hydroxyestrogen (4-OH-E2), 4-hydroxyestrone (4-OH-E1) or estrogen’s quinone derivatives. The process is viciously regulated by the CYP1B1 activity that controls E2 metabolism and concurrent interaction of products thereof with cigarette smoke carcinogens which, on further transformation, aggravate the ROS formation. Catalyzing the hydroxylation at 2 and 4 positions of E1 and E2, the generated 4-hydroxylated metabolites are carcinogenic [22,23]. Before being formed at once, the endogenous catechol estrogens can be oxidized by any enzyme having an oxidative ability, generating the vulnerable electrophilic estrogen o-quinones and semiquinones. These quinone derivatives aggravate ROS formation through a series of redox cycling events and are detrimental to cells in multiple manners. For instance, o-quinone metabolism indirectly generates free hydroxyl radicals, the most harmful oxidizing agents. These molecules induce DNA damage via multiple mechanisms, such as inducing single-strand breaks, chromosomal aberrations and 8-oxo-2′-deoxyguanosine formation. The quinones and semiquinones can also inflict direct cellular DNA damage by forming adducts, culminating in genotoxic effects (depurination). Studies have highlighted the capability of catechol estrogens, o-quinones or their metabolites to bind to ERs and, on further transportation to the ERE in the nucleus, to result in DNA mutation via free radical emission [24,25]. A prospect of further caution pertains to overexpressed CYP1B1 which is enacted through long-lasting tobacco combustion. Thereby, the risk of concurrent free radical and ROS-induced damage is higher in chain smokers or those with a long smoking association. Estrogens have important roles in lung carcinogenesis and subsequent complications of lung cancers [26,27,28].

Of note, NSCLC accounts for about ~85% of all lung cancers [29] with an abysmal (13–18%) five-year survival rate [30,31]. Contrary to male gonads (testes), female gonads (ovaries) produce higher levels of aromatase, consequently having higher estrogen amounts in the blood [32]. Thus, females have a greater risk of being affected by various estrogen-dependent complications, including cancers [33,34]. Studies reveal a more than twice higher adenocarcinoma risk in smoking females advised or undergoing estrogen replacement therapy (ERT) than those not subjected to ERT, odds proportion being 32.4 to 13.1. Contrary to this, non-smoking women taking ERT exhibited no significant adenocarcinoma risk [35]. Lately, though, the impact of HRT on a possible lung cancer manifestation has revealed a 50% higher risk for women under a combined hormone therapy (estrogen with progestin) [36,37,38,39]. Besides increased lung cancer risk, the HRT mediated via estradiol and progesterone combined also exhibited a significant association between a younger median age for lung cancer diagnosis and shorter median survival duration [39,40,41]. Apart from CYP1B1, the CYP1A1 gene encodes for phase I enzyme which metabolizes polycyclic aromatic hydrocarbons (PAHs) in tobacco smokes, preventing the precarcinogen from turning carcinogenic [42]. The circulating steroid hormones in females impair this CYP1A1 action due to their interaction with receptors in the lungs of sufferers. As a result, the carcinogen formation becomes unregulated which aggravates the LC risk. Studies in animals support the above predictions with female mice being more sensitive to chemically induced lung carcinogenesis, which tends to be inhibited by ovariectomy [43]. In a study monitoring the implicit activities, Hammoud and colleagues examined the lung tumor intensity subsequent to estradiol intake (2 μg per day for 10 weeks). While tumor count and volume declined after ovariectomy contrary to intact female mice, estradiol administration increased the tumor count and volume in the ovariectomized mice compared to untreated female mice. Similar distinctions were observed in male mice administered equal E2 concentration [44].

Estrogens work through estrogen receptors (ERs)—ERα and ERβ—either through genomic or nongenomic mechanisms [45,46]. Using various ER knockout mice models, Couse and colleagues described the importance of ERs in health and physiology [47]. The details of the ER mechanism of action including their genomic and non-genomic activation could be traced in multiple sources [3,48]. Physiologically, ERs are expressed in normal lung tissues as well as in NSCLC and respond transcriptionally to 17-β-estradiol or E2 [49,50]. Besides ERα and ERβ, other estrogen receptors implicated in the severity of lung cancer are membrane-bound G-protein-coupled estrogen receptors (GPERs). Generally, GPERs are transmembrane proteins. Under physiological conditions, E2–GPER interaction is involved in various rapid non-genomic actions [51,52]. A high-level GPER expression was observed in human and mice lung cancer tissues and in the lung cancer cell lines. In these cells, the receptors are predominantly located in cytoplasmic cell organelles, such as the Golgi apparatus and endoplasmic reticulum [53,54]. In addition, under NSCLC aggravated state, cytoplasmic GPER expression was found as being correlated with more advanced cancer stages (IIIA-IV), lymph node metastasis and poor differentiation [53]. A number of progrowth and pro-proliferative pathways are activated by the ligand-bound GPER and thereby activate tumor/cancer progression [55,56,57,58].

Besides ERs (ERα and ERβ) and GPERs, another receptor closely related to estrogen-dependent signaling is the oncogenic protein epidermal growth factor receptor (EGFR). In NSCLC complication, ~89% of patients exhibit an EGFR overexpression or mutation. Thus, the EGFR family of tyrosine kinase receptors is one of the most important signaling molecules in NSCLC complication [59]. EGFRs are colocalized with ERβ in breast cancer and NSCLC cells [60]. In NSCLC, E2 activates the EGFR pathway promoting cell proliferation, survival, angiogenesis, migration and metastases [61]. Additionally, the orphan nuclear receptors, i.e., estrogen-related receptors (ERRs), which have considerable structural and functional similarities with ERs, are implicated in breast cancer complications [62], alongside aggravated tamoxifen resistance in breast cancer cells [63]. Of note, tamoxifen is a selective estrogen receptor modulator (SERMs). ERRs are detected in the lung tissues both in normal [64] as well as NSCLC-affected lung tissues, aggravating the tumorigenesis [65,66].

A large number of in vitro cell culture and animal model experiments including knockout mice models were conducted to examine the efficacy of anti-estrogen therapies against NSCLCs. In these studies, the efficacy of impaired in vivo estrogen synthesis by aromatase inhibitors was examined. Additionally, SERMs such as tamoxifen or selective estrogen receptor downregulators (SERDs) such as fulvestrant were utilized either alone or in combinations of anti-EGFR molecules [67,68,69]. Prossnitz and colleagues discussed the importance of using pharmacological GPER modulators [70]. ERRs were also targeted because of their considerable structural and functional similarities with the ERs [65,66]. Recently, a number of clinical trials have been completed by targeting estrogens and associated molecules against NSCLCs [71,72,73]. This comprehensive review article sheds light on the role of endogenous and exogenous estrogens as well as ERs, GPERs, EGFRs and ERRs in the NSCLC complication and the efficacy of targeting these molecules in the NSCLC subjects.

## 2. Exposure of Mammalian Lungs to Endogenous and Exogenous Estrogens Including Synthetic Estrogens, Phytoestrogens and Xenoestrogens

Under thoroughly normal physiological conditions, females express high levels of aromatase (CYP19A1) enzymes from the granulosa cells of Graafian follicles within the ovaries. In the process of steroidogenesis, aromatases catalyze testosterone-to-estrogen conversion. Thus, in complete normal physiological conditions, a female body by itself (naturally) contains a large amount of estrogens with a proportionately low level of testosterone. These naturally produced estrogens have physiological roles both in gonadal and nongonadal tissues. In comparison, the Leydig cells of testes (in males) synthesize a low level of estrogens owing to a weaker aromatase activity, resulting in concurrent testosterone accumulation along with its subsequent conversion to dihydrotestosterone (dHT) through the action of 5-α-reductase [6,7,32]. Estrogens once produced from the gonadal tissues (ovary/testis) are released into the circulation, ultimately reaching various body parts including lungs. Besides major synthesis in the gonads, a low level of aromatases is also locally expressed in nongonadal tissues which are capable of synthesizing estrogens that complicate the cancerous progressions [74]. The 17-β estradiol (E2) is the major estrogen present in the female circulation with a half-life of 3.5 h [10], which is metabolized to estrone (E1) and estriol (E3) within the liver. Collectively, E1, E2 and E3 are the major backbone of various estrogens expressed in the mammalian body. Some other estrogen-active metabolites produced in the mammalian body include catechol estrogens (2-OH-E-2 and 4-OH-E-2), which are further metabolized into highly reactive estrogen quinones (E2-2,3-Q and E2-3,4-Q) [11,75,76]. Some of the estrogen metabolites such as 4-OH-E2, a product of the CYP1B1 enzyme, exhibit mutagenic and carcinogenic effects [77]. The 17β-estradiol-to-4-OH-E2 metabolic conversion is induced via cigarette smoke exposure. In various tissues including lungs, estrogen metabolites regulate the mutation and subsequent proto-oncogenes-to-oncogenes conversion which increases the lung cancer risk [78]. The development of lung cancer in humans may be associated with gene deletions on the chromosomes 1, 3, 11, 13 and 17. Prominent proto-oncogenes induced by these genetic aberrations include c-jun, ras and c-raf1 along with a loss of tumor suppressor gene p53. Amongst these, c-erbB-2, c-sis and c-fes are prominently expressed or missing in the NSCLC and can be of substantial importance in the selection of a differentiation pathway. Preclinical studies have revealed major NSCLC proliferative activities regulated through genomic and indirect non-genomic mechanisms of E2. The c-myc oncogene is often noted as being amplified in small-cell lung cancer (SCLC) cell lines. Investigations demonstrate the ER influence on NSCLC cells mediated via EGFR-signaling-driven cell-cycle regulation, the cAMP, MAPK and AKT pathways and the promotion of c-myc and cyclin D expressions [79,80]. Despite being well-known for estrogen induction, the c-myc promoter does not possess any estrogen-response elements (EREs).

While exact investigations tracing estrogen response in NSCLC are rather too scarce, the results from the modulated expressions in other tumors (primarily breast) indeed offer a basis of functional link. A 2011 study of this kind discussed the experience of distant elements being involved in estrogen-induced gene expression. Analysis revealed an insignificant effect on c-myc proximal promoter activity, though it stimulated the activity of a luciferase reporter characterized by a distal 67 kb enhancer. This activity of estrogen was noted as dependent on a half-estrogen-response element apart from an activator protein (AP-1) site residing within this enhancer. The conservation of this AP-1 binding site in 11 distinct mammalian species and active estrogen-AP-1 jointly suggests AP-1 as the source to propagate the tumor development. Exclusive involvement of AP-1 in this tumor-promoting activity was confirmed by small interfering RNA experiments and chromatin immunoprecipitation assays wherein estrogen receptor-AP-1 cross-talk was demonstrated as an essential factor to induce c-myc expression [81]. Noted human lung cancer cell lines are characterized by the generation of multiple growth factors that are engaged in proliferation via paracrine and autocrine loops through specific receptors. The resultant products from some activated proto-oncogenes (c-sis and c-erbB-2) are the homologous sequences to some specific growth factor (such as platelet-derived growth factor (PDGF)) and the EGFR, commonly identified in lung cancer. Actions of these growth factors serve as decisive links for tumor progression via intracellular signal transduction and specific oncogenic activation. The observations of a 1990 study mention the presence of fur gene in 32 of the 40 NSCLC examined biopsies [82]. The fur gene encodes for a membrane-associated protein, and its involvement in NSCLC development is due to its location immediately upstream of the fes/fps proto-oncogene. With its receptor-like attributes, the fur gene together with fes/fps proto-oncogene exhibits a tyrosine protein kinase activity, which draws significance from the therapeutic rationale of using selective tyrosine kinase inhibitors of EGFRs, whose overexpression remains a recognized hallmark of NSCLC [83]. The above 1990 investigation examined c-erbB-2 expression in 60 patients (SCLC and NSCLC both) along with 11 lung cancer cell lines. The product from c-erbB-2 is a membrane protein having tyrosine kinase activity with c-erbB-2 having a sequence homology with EGF receptor. The gene for c-erbB-2 was mapped to chromosome 17 (17q21) wherein only two of the 60 examined biopsies exhibited amplification. Interestingly, all the seven investigated NSCLC cell lines showed an increased c-erbB-2 gene expression with a characteristically high transcription extent compared to the SCLC cell lines. With a sequence homology of EGF receptor, for which the estrogenic activity is well-known for aggravating tumorigenesis via EGFR signaling, the c-erbB-2 gene presents a high likelihood of being intercepted by estrogen for an enhanced carcinogenic response. A different study noticed enhanced c-erbB-2 proto-oncogene expression in squamous and large-cell undifferentiated cancers contrary to the adenocarcinomas. Encoding a growth factor receptor on glandular epithelium, this gene is implicitly amplified only in an adenocarcinoma. Abnormal expression of this gene is noticed at a higher rate in advanced than early-stage cancers [84]. This investigation studied the expressions of c-myc, c-myb, c-ras-Ha, c-erbB-1 and c-erb-B-2 proto-oncogenes in NSCLC and observed their detectable abnormalities in adenocarcinomas (ten out of sixteen) and large-cell cancers (two out of two).

The c-sis oncogene in the same study [83] was noticed as prevailing homologous to platelet-derived growth factor (B-chain), expressed in five of the six examined NSCLC cell lines. Transcripts for PDGF A-chain were detected in all studied six NSCLC cell lines. The transforming growth factors (TGFs) α and β were positive in four and five NSCLC cell lines. Injection of studied NSCLC cell lines into nude mice revealed varying extents of collagen-rich tumor stromata, suggesting a paracrine functioning of PDGF and TGF in NSCLC cell lines. Here again, five of the studied six NSCLC cell lines possessed EGF receptor transcripts. Thereby, a significant association of c-erbB-2, c-sis and c-fes with EGFR expression infers an aggravated NSCLC carcinogenesis via ER–EGFR interactions. Thus, the NSCLC manifestation is the outcome of either the transcription factor activation followed by translocation to nucleus (activation) or via direct binding with an ERE-promoter-sensitive locus of estrogen-responsive genes (ERG).

Besides natural estrogens, females may be additionally exposed to exogenous synthetic estrogens such as ethinyl estradiol (EE) [12,13,85] or equine estrogens [86]. The human body, in particular female, is exposed to these estrogens either as contraceptive pill constituents or as the components of hormone response therapy (HRT). Diethylstilbestrol (DES) is another synthetic estrogen used in various scientific experiments to monitor the effect of estrogens on the mammalian body [87]. Both EE and DES have carcinogenic potentials. Other synthetic estrogens used by human subjects are estradiol valerate, estropipate, conjugate esterified estrogen and quinestrol. Additionally, the human body may be exposed to a number of phytoestrogens such as genistein as and when consumed through certain plant-derived foods [16]. Some phytoestrogens are considered endocrine disruptors, resulting in significant adverse health effects [20]. Bisphenol-A (BPA, a constituent of various plastic containers including drinking water bottles) is another exposure form of exogenous compounds that acts as an endocrine disruptor and promotes the migration and invasion of lung cancer cells [88]. Numerous studies in the Chinese population showed significantly higher BPA levels in NSCLC patients than in healthy controls, and therefore, BPA exposure is considered an important NSCLC complication risk factor [89].

A large group of environmental agents or pollutants comprise xenoestrogens, exerting estrogen-like effects together acting as endocrine disruptors. Environmental pollutants such as arsenic, pesticides, fungicides, polycyclic aromatic hydrocarbons (PAH), dichlorodiphenyl-trichlorethane (DDT), polychlorinated biphenyl (PCB), heavy metals and other pollutants such as phthalates, alkylphenols and certain drugs (e.g., antiepileptic drugs) may alter the level of estrogens in the blood/tissues or bind to ERs and alter their activities [90,91,92,93,94,95]. For example, a comparative study of unexposed versus transplacental exposed arsenic through drinking water in the fetal lungs of female mice showed a significant enhancement in the ERα expression within the lungs [96,97]. Additional studies have shown that alcohol drinking increases estrogen synthesis alongside the chances of cancers [98,99]. Other potential xenoestrogen molecules such as volcano dust, forest fire smoke, air pollution originated from coal-fired power stations and chemical industry pollutants, automobile exhaust, various cosmetic products, cigarette smoke constituents and overcooked foods such as grilled meat affect the endocrine system. However, smoking remains the major cause of lung cancer in humans [100]. Cigarette smoke contains methylnitrosamino-pyridyl-butanone, a compound that acts as a powerful carcinogenic agent and is responsible for ERβ activation [101]. Cotinine, a nicotine metabolite, inhibits aromatase [102] and therefore acts in a gender-specific manner [103]. Figure 2 portrays the methylnitrosamino-pyridyl-butanone and cotinine chemical structures, wherein pi-conjugation in the N-substituted phenyl ring and double-bonded oxygen in close proximity of nitrogen are the common features and could have a discrete involvement in the toxicity induction. Knowledge of such structural inputs could aid in moderating the localized toxicity of each of these compounds with the design of accurate scavenging or neutralizing moieties.

By inhibiting aromatization of testosterone to estrogens, nicotine/cotinine aggravates the pulmonary tissue level of testosterone. Polonium 210 in cigarettes may have similar activity as other metalloestrogens [104]. Since cigarette smoke produces a few hundred chemicals, it is essential to evaluate pro- or anti-estrogenic activities of each of these compounds. Comprehensive review articles by Fucic A and colleagues in 2010 and 2012, respectively, describe the role of various environmental agents having xenoestrogenicand potential carcinogenic activities [17,105]. Thus, unlike normal physiological conditions, in NSCLC complicated pathophysiology, mammalian lungs are exposed to estrogens. A number of studies claimed that lung cancer cells produce their own estrogens and aromatase inhibitors may have beneficial effects against lung cancers including NSCLCs [106,107]. Expression of ERs and aromatases may have prognostic value in NSCLC severity [108]. Besides endogenous factors, the various exogenous factors such as synthetic estrogens, phytoestrogens, certain environmental factors and xenoestrogens affect the expression level of estrogens as well as of estrogen-related various receptors and therefore affect NSCLC manifestations.

## 3. Expression of Estrogen Receptors by Mammalian Lungs and Their Physiological Roles

Broadly, ERs are divided into two types—ER-alpha (ERα) and ER-beta (ERβ)—each having various splice variants and tissue-specific expression. These ER variants are generated either by alternative splicing, initiation of translation [109,110] or proteolysis [111]. It was Kupier and groupmates who for the first time characterized the tissue-specific distribution of ERα and ERβ transcripts in rats and claimed ERβ as the predominant isoform prevailing in the lungs [112]. Later on, other studies also detected both ERs in the lung tissues and cell lines [49,113]. In healthy lung tissues, pneumocytes and bronchial epithelial cells are the predominant sites of ER expression, and sex steroids have effects both on the normal physiology and diseased conditions [8,9]. The study by Ivanova and colleagues ascertained the ER subcellular localization including that of cytoplasm, nucleus, etc., in normal bronchial epithelial cells as well as NSCLC cells. Their results confirm the prior observations of Kupier and associates that bronchial epithelial cells express nearly twice as much ERβ than ERα [50,114]. ER knockout mice models were used to understand the importance of ERα, ERβ or both in the overall phenotype as well as the development and function of lungs. Interestingly, ERα null mice showed no phenotypic alteration in the lungs [47]. However, working on ERβ knockout mice, Patrone C and accomplices observed ERβ as the regulator of lung alveoli population, surfactant protein production and overall homeostasis of the lung tissues [115]. The results of this study indicate that the lungs of ERβ knockout female mice exhibited reduced surfactant protein, platelet-derived growth factor A (PDGF-A) and granulocyte macrophage colony-stimulating factor (GM-CSF) expressions. Another study by Morani and colleagues revealed that at 5 months of development, both male and female ERβ-deficient adult mice exhibited abnormal extracellular matrix deposition and alveolar collapse, together resulting in systemic hypoxia [116]. This observation further suggests that ERβ is involved in the structural and functional development of lungs in both adult male and female mice, in a ligand-independent manner, or that estrogens act via a local autocrine mechanism. In another attempt, Ciana and colleagues used 26 different tissues, including those of lungs from transgenic and ovariectomized female mice models for ERE-luciferase reporter assay activities against 17β-estradiol-ERs stimulation. Significantly, their results show that lungs in these mice models displayed 17β-estradiol-ER-dependent ~15-fold induction of luciferase reporter gene activity. This study identified lungs as the most important target tissues of the mice model for the estrogen-mediated cellular actions [117]. In a similar ERE-luciferase reporter gene assay in both male and female mouse models, Lemmen and teammates also identified lungs as an important target site for 17β-estradiol activities [118]. Of note, under completely normal physiological conditions, 17β-estradiol/E2 is the major estrogen present in females. The likelihood of ERβ playing more important and decisive control activities in lungs contrary to the ERα is further strengthened by the microarray data, revealing ERβ tumor expression as linked with nearly 500 gene variations contrary to a mere 20 for ERα [119]. All these studies collectively indicate that estrogens and ERs, particularly ERβ, not only play multiple important roles toward maintaining a healthy lung physiology but are also associated more strongly with LC intracellular transformations.

## 4. Role of Estrogen Receptors in Non-Small Cell Lung Cancer Complication

To implicate ERs in the complications of NSCLCs, it is essential to determine that the NSCLC tissues are exposed to estrogens either through localized enhanced aromatase expression or via sufficient circulatory estrogens being synthesized and secreted in the gonads. Exposure of the human body to exogenous estrogens is also possible either through administration of synthetic estrogens/phytoestrogens orxenoestrogenic agents. Since both genomic or non-genomic actions of estrogens involve interactions with ERs or other associated receptors (e.g., GPER/EGFR), detecting the expression level of these receptors in the NSCLC tissues may be crucial. Indeed, aromatase activities have been detected in NSCLC cell lines as well as ~86% of tumors [108]. A number of studies including one by the Mollerup group reported the ER prevalence in lung tumors and NSCLC cell lines [113]. Additionally, a relationship of the hormonal status in the cancer-affected tissues with the expression index of ERs has been established. Of note, it is well known that premenopausal women generate large contingents of estrogens as compared to postmenopausal women and men of all age groups. A comparative analysis of ER expression extent in cancer patients, vis-à-vis age/gender group of pre- and postmenopausal women and men, revealed the highest ER expression intensity in the premenopausal women. In this study, the men cancer patients exhibited a minimal ER expression, suggesting a critical role of either circulating or local tissue level of estrogens in the ER expression [120]. Additionally, overexpression of ERs is detected in lung adenocarcinoma. Of the two ER types, the ERβ is more abundant in lung cancer. Studies have reported an ERβ overexpression in 60–80% of tumors, irrespective of gender [121]. In a noted effort, Kawai and associates explained the differential ERα detection pattern in NSCLC samples using an ERα antibody (Ab) raised against either a full-length or an N-terminus or a C-terminus ERα protein. The Ab against the C-terminus ERα region detected a predominantly cytoplasm-localized protein compared to the one raised against the N-terminus of the protein. The investigators predicted that in NSCLC, ERα is N-terminal-deleted and lacks the nuclear localization signal [122]. In a similar study on the prevalence of ERα splice variants, ERα36 was reported in NSCLC specimens, while wild-type ERα was minimally expressed. The results of this study claim that in normal lungs, the wild-type ERα is quasi-absent [123]. In comparison, both in normal physiology as well as in cultured NSCLC cells, the ERβ is predominantly localized within the nucleus [49,122]. A number of studies including that of Baik and colleagues described the ERβ nuclear localization as a more reliable prognosis factor contrary to that of the cytoplasmic ERβ expression [27]. Studies also claimed that the cytoplasmic and the nuclear ERβ co-expression were correlated with a low survival rate compared to the one without co-expression [124]. The ERs are also localized in the plasma membrane [52,125]. Studies by Gao and teammates claimed the ER involvement in NSCLC progression by modulating the membrane receptor signaling network [126]. It is predicted that the localization of ERs in the specific cellular compartment including plasma membrane, cytoplasm and/or nucleus may have a distinct function and affect the prognosis differentially via a genomic or non-genomic pathway. The ERs expressed by lung bronchial epithelial as well as NSCLC cells respond transcriptionally to E2/17β-estradiol [66,67,114]. In a Kras-activated and p53-deletion-induced lung adenocarcinoma mice model, administration of E2/17β-estradiol promoted the tumor progression. In this experimental mice model, male and ovariectomized female mice respond likewise in response to17β-estradiol administration [44]. The poorer clinical outcomes observed in NSCLC patients may be related to the proliferative as well as survival responses to 17β-estradiol. A number of studies documented the specific cell signaling molecules and associated pathways involved in the 17β-estradiol dependent proliferation, survival and growth of in vitro cultured NSCLC cell lines and tumor xenografts [60,67,121,127]. However, there is a lot of controversy regarding beneficial versus adverse effects of HRT in the human body [14,15]. Similarly, the role of ERs in NSCLC remains controversial, and the mechanisms of action of ERs in NSCLC complications are not conclusive. There is no consensus on whether ERβ expression plays a role in survival. Some studies have suggested a protective effect of ERβ nuclear expression [128,129,130], which may only be significant in men [130]. It is possible that the presence of nuclear ERβ confers a hormonal dependence for growth rather than on the other more aggressive oncogenic pathways, leading to a comparatively better survival. A number of comprehensive review articles described various aspects of ERs in the NSCLC complication [55,131].

## 5. Efficacy of Anti-Estrogen and Anti-Estrogen Receptor Molecules against Non-Small Cell Lung Cancers

Aromatase inhibitors that include irreversible steroid inhibitors/non-steroidal inhibitors (e.g., anastrozole, letrozole, etc.) and selective ER modulators (SERMs, e.g., tamoxifen, a competitive inhibitor with equal ERα and ERβ affinity) or selective ER downregulators (SERDs, e.g., fulvestrant/ICI182780, a pure anti-ER molecule with equal affinity for ERα and ERβ which degrades both ERα and ERβ) are used against estrogen/ER-dependent cancers. The efficacy of these molecules against NSCLC complication was examined by in vitro cell culture and animal model experiments including xenograft and ER knockout mice models and clinical studies involving NSCLC patients. Details of the estrogen activities in lung cancer complication, particularly in women, have been recently reviewed [132,133]. In cell culture experiments using A549 NSCLC cells, it was shown that lowering the estrogen concentration inhibited the A549 cell’s growth [134]. Another significant in vitro study revealed that aromatase inhibitors such as letrozole significantly decreased cell proliferation while exemestane reduced tumor growth and increased cell apoptosis, alongside inhibiting cell migration and invasion [135]. The efficacy of the SERD molecules such as fulvestrant was also examined in the in vitro cell culture based experiments and tumor xenograft mouse model. The results of these studies show significant inhibition of 17β-estradiol-induced NSCLC (cell lines) grown in culture and as tumor xenografts. In this study, the combined activity of anti-ER and anti-EGFR molecules emerged more efficacious pertaining to the anti-NSCLC activities [67]. A number of similar experimental studies focused on the efficiency of fulvestrant in NSCLC suppression [106,136,137]. Data presented by Wang and colleagues showed that dexamethasone suppresses the growth of NSCLC cell line A549 injected xenograft tumors via inducing estrogen sulfotransferases and decreasing estradiol levels in tumor tissues, suggesting that dexamethasone may be used as an anti-estrogenic agent for the NSCLC treatment [138]. Clinical observational studies consisting of 6500 breast cancer survivors subjected to anti-estrogen therapy showed lower subsequent lung cancer mortality [139]. In another similar study consisting of 2320 patients, a strong anti-estrogen therapy and lung cancer motility correlation was observed [140]. More recently, three separate and independent clinical trials have been conducted to evaluate the anti-estrogen/anti-ER therapeutic efficacy toward lung cancer [71,72,73]. Based on the results, it is predicted that, in the future, ER detection in NSCLC could be considered reliable treatment recourse for women. Comprehensive review articles discussed the estrogens and their receptors in influencing the development as well as microenvironment of lung cancers and the significance of clinical studies exploring the anti-estrogen/anti-ER therapy usefulness against lung cancers (Table 2) [141,142].

## 6. G-Protein-Coupled Estrogen Receptors and Non-Small Cell Lung Cancers

In 1997, Carmeci and colleagues discovered an estrogen-bound transmembrane receptor in breast cancer tissue [151]. In the same year, Takada and colleagues [152] and, subsequently in 2005, Revankar and group [153] cloned the G-protein-coupled estrogen receptor cDNA sequence using differential cDNA library analysis of the human breast adenocarcinoma cell lines, MCF-7 and MDA-MB-231. In 2007, the International Union of Basic and Clinical Pharmacology officially named this molecule a G-protein-coupled estrogen receptor (GPER, also known as GPER1 or GPR30) [154]. Chromosomal mapping shows the human GPER gene being located on chromosome 7, containing a 1128 bp open reading frame that encodes a 375-amino acid receptor protein [151,155]. GPER is a seven-transmembrane-bound ER [44]. The presence of GPER in the cell surface or plasma membrane is supported by the experimental studies of Filardo and associates [52,156]. Using HEK-293 or SKB3 breast cancer cells, the investigators showed that the cell-impermeable E2-protein conjugates, E2-BSA and E2-horseradish peroxidase, promote GPR30-dependent increment in the intracellular cAMP concentrations. To further confirm that the plasma membrane is the site of GPR30 action, the investigators completed the binding assay using [3H]17β-E2, followed by subcellular fractionation. Their results confirm that plasma membrane isa site of GPR30 actions with [3H]17β-E2 binding. Thus, in this study, G-protein activation was revealed as being associated with plasma membrane but not with microsomal or other fractions prepared from HEK-293 cells or SKBR3 breast cancer cells [156]. GPER is also localized in the cytoplasmic cell organelles such as the Golgi apparatus and endoplasmic reticulum [156,157,158,159].

GPER plays a critical role in regulating several normal physiological functions mediated by estrogens as evidenced by its tenfold greater capacity of estrogen binding compared to ERα [51]. Supporting experiments by Filardo and colleagues confirmed a GPER involvement in rapid non-genomic actions [52]. Using breast cancer cell lines, it was shown that estrogens have the ability to directly bind with GPR30 without ERα and ERβ involvement, alongside activating the ERK1/2 and mitogen-activated protein kinase (MAPK) pathways involved in the proliferation of various cells [159]. Of note, an uninterrupted or stringent MAPK pathway is involved in aggravating multiple pathophysiological conditions, including cancers. Later on, Thomas and associates also confirmed estrogen binding with plasma-membrane-bound GPER, followed by their concurrent non-genomic activation of various downstream signaling pathways [125]. The functional mechanism of GPER differs from those of typical nuclear receptors such as ERα and ERβ. Some of the molecules and cell signaling pathways activated by GPER are adenylyl cyclase, cAMP, calcium mobilization and MAPK. Other cell signaling pathways activated by GPER include EGFR/MARL transactivation, the Hippo/yes-associated protein 1 (YAP)/transcriptional coactivator with PDZ-binding domain (TAZ) pathway and phosphoinositide 3-kinase (PI3K)/Akt pathway (Figure 3) [58,70]. A comprehensive discussion based on the studies conducted on GPER knockout mice confirmed the perception that GPER is in fact an ER with specific physiological functions mediated through nongenomic actions [70]. In 2018, Barton and colleagues described the major discoveries and events pertaining to the GPER functioning through a comprehensive review article [160].

GPERs are expressed in the lung tissues that include normal bronchial epithelial cells [102]. In complete physiological conditions, various human tissues including the lungs are subjected to E2–GPER non-genomic actions [161]. A high-level GPER expression was observed in the human and mice lung cancer tissue and in the lung cancer cell lines. In these cells, the receptors are predominantly located in the cytoplasm [54]. In lung cancer cells, high-level GPER expression is associated with high ERβ expression [53]. A number of progrowth, pro-proliferative pathways are activated by the ligand-bound GPER, in turn activating tumor/cancer progression. Metastasis is higher in cells with GPER expression in large-cell carcinoma than squamous-cell carcinoma or adenocarcinoma [54], ably supported by the reduced extent of lung metastasis in GPER knockout (KO) mice. Multiple studies claimed that GPER acts as a modulator of neoplastic transformation [56]. In the lung cancer cells, the GPERs are linked to various MMP activation and therefore assumed to be involved in cancer cell migration [88]. In addition, cytoplasmic GPER expression was found as implicated in the manifestation of advanced cancer stages (IIIA–IV), lymph node metastasis and poor differentiation in NSCLCs [158]. Studies by Avino and associates claimed that GPER-driven activation of insulin-like growth factor-I (IGF-I)/IGF-IR via ERK, p38 and Akt stimulation leads to migration of cancer cells [162]. A recent investigation by Shen and colleagues revealed an oncogenic activity of GPER on NSCLC cells via regulation of YAP1/QKI/circNOTCH1/m6A methylated NOTCH1 mRNA signaling [58]. In contrast to these studies, some other investigations claimed that GPER activation inhibited migration of human NSCLC cells via suppressed IKK-β/NF-κB signaling pathways [57].

## 7. The Agonists and Antagonists of G-Protein-Coupled Estrogen Receptors and Potential Non-Small Cell Lung Cancer Therapy

GPERs interact with endogenous natural estrogens and their various in vivo metabolites along with different phytoestrogens and xenoestrogens. However, while some of these molecules act as GPER agonists, others act as antagonists. For example, while 17β-estradiol (E2), the major estrogen in female blood, is a predominant ligand of GPER with agonistic activities [163], estrone (E1) and estriol (E3) (the major in vivo 17β-estradiol (E2) natural metabolites) act as antagonists [164]. ICI182780 (fulvestrant), an SERD [165], was likewise reported to be a GPER agonist [166] with a 30–50nM GPER binding affinity [124]. Similarly, tamoxifen, an SERM, has been reported to act as a GPER agonist in SKBR3 cells [167]. Even the E2 metabolite, the catechol estrogen 2-OH-E2, binds GPER with a high affinity and acts as a GPER antagonist [168]. Additionally, various studies examined the phytoestrogen and xenoestrogen effects on GPERs. Phytoestrogens such as quercetin, genistein, isoflavones, flavones, coumestans, lignans, saponins and stilbenes and synthetic estrogenic compounds in pesticides, herbicides and some plastic monomers including polychlorinated biphenyls and dioxins, bisphenols, methoxychlor and alkylphenols activate GPERs [169,170]. The xenoestrogens, BPA and dichlorodiphenyltrichloroethane (DDT) exhibit a relatively lower affinity toward GPER (at ~0.6 mM) contrary to the (2.8–10) mM range for DDT (albeit it also depends on implicit isomer structure) [171]. Figure 4 summarizes the possible estrogen-targeted therapies that could be exploited to arrest the NSCLC tumorigenesis. The mechanisms relate to the inhibition of estrogen action by the competitive inhibitors of estrogen receptors (e.g., tamoxifen), degradation of estrogen receptors (e.g., fulvestrant or ICI182780) or inhibition of estrogen synthesis by aromatase inhibitors (e.g., anastrozole). Other targets of inhibition are GPER (e.g., by G15, G36) or EGFR by EGFR inhibitors (e.g., gefitinib, erlotinib). A number of molecules have been synthesized that exhibit agonistic or antagonistic potentials against GPERs. The most important of these are G-1, G-15 and G-36. Figure 5 depicts the chemical structures of G-1, G-15 and G-36 GPER antagonists, distinguished in terms of alkoxy, null and extended alkyl chain substitutions at the ortho positions of the terminal phenyl ring. A common prospect for designing more potent analogs inferred from these structures is the presence of nitrogen and hydrogen together (intramolecular hydrogen bonding), bromine as a substituent, in and out of the plane symmetries of hydrogen atoms, distributed pi-conjugation (hydrophobic receptivity) and five-membered ring having two oxygens (controlling oxidative balance).

In 2006, Bologa and colleagues identified the GPER-selective agonist G-1 [172,173]. The antagonist G-15 was discovered in 2015 and exhibits a 0.5-fold stronger binding compared to G-1 along with a 1000-fold selectivity toward ERα and ERβ [174]. Another GPER antagonist, G-36, was synthesized by Dennis and associates in 2011. Compared to G15, G-36 has been in wide use for GPER study owing to its lower off-target activity and weaker cross-reactivity with ERα [175]. The binding affinity of various agonists and antagonists varies, and therefore specific concentration being used in experiments is highly diligent. Recently, Liu and colleagues showed that GPER antagonist G15 decreases estrogen-induced development of NSCLCs [53]. This study concluded that GPER may have the potential to be used as a therapeutic target against NSCLC manifestation. Toward this direction, two separate comprehensive review articles by Xu and Rouhimoghadam groups described the potential of various GPER agonists or antagonists as therapeutic agents against various diseases [176,177]. More clinical studies are necessary to evaluate the anti-NSCLC GPER regulation.

## 8. Interactions of Estrogen Receptors and Epidermal Growth Factor Receptors for Aggravated Tumorigenesis in Non-Small Cell Lung Cancers

The oncoprotein epidermal growth factor receptors (EGFRs) are a family of tyrosine kinase receptors. In NSCLC manifestation, EGFR-mediated signaling remains one of the most important pathways since 89% of NSCLC patients exhibit either EGFR overexpression or mutation [59]. The EGFR is known to be involved in NSCLC growth, protection against apoptotic defense and regulation of angiogenesis [67]. In NSCLC, E2 activates the EGFR pathway promoting cell proliferation, survival, angiogenesis, cell migration and metastasis (Figure 6) [61]. A number of studies claimed that considerable EGFR and ER cross-talk prevails in NSCLC pathogenesis. In 87.5% of the ER-positive lung tumors, serine residues in the ER molecules phosphorylate the EGFR although ligand involvement in this cross-talk was not confirmed [26,127]. However, other studies claimed that estrogens bind with membrane-bound ERs to activate the cytoplasmic signaling pathways including that of EGFR signaling pathways [178]. Membrane ERs were also found to be co-localized with EGFR in lung tumors [60].

In the in vitro cultured NSCLC cell lines, the EGFR expression was downregulated in response to estrogen and upregulated in response to anti-estrogens, indicating EGFR expression upregulated in the absence of estrogens. Conversely, the ERβ expression is increased in response to gefitinib, an anti-EGFR molecule, and decreased in response to EGF [67]. In lung adenocarcinoma, a strong association between the ERs and EGFR mutations has been reported [179,180]. These studies provided experimental evidence that in lung cancer tissues, the ER and EGFR pathways functionally interact with each other, thereby augmenting well for the rationale to use the combined therapy [69]. Further in vitro studies indicate that during lung carcinogenesis, the ER and EGFR interact or cross-talk with each other. Stimulation of ER has been implicated to decrease the activity of the EGFR signal, which in turn increases the activity of ER [181]. The association between ER expression and clinico-pathological parameters such as EGFR gene mutation may elucidate the prognostic role of ER expression in lung adenocarcinoma. Studies by Nose and colleagues demonstrated that strong ERβ nuclear expression was implicated in the manifested EGFR mutations, in turn resulting in its favorable prognostic significance being influenced by EGFR mutations in lung adenocarcinoma patients [182]. Additionally, similar studies by Deng and colleagues revealed an association of ERβ expression with EGFR mutations [179]. Following EGFR signaling interception by estrogens, several downstream signaling pathways such as MEK/ERK and PI3K/AKT were activated which promote lung cancer metastasis via epithelial–mesenchymal transition [183].

## 9. Potentials of Dual Targeting of Estrogen Receptors and Epidermal Growth Factor Receptors against Non-Small Cell Lung Cancers

Overexpression of EGFR is detectable in approximately 80–85% of NSCLC patients. Selective inhibition of EGFR tyrosine kinase activities has a significant clinical impact on therapy in lung cancer patients [83]. Anti-ER and EGFR molecules are used for dual-mode-strengthened lung cancer therapy. In NSCLC, the effect of combined anti-estrogen and TKI therapy has been investigated due to the functional relationship between estrogen and EGFR pathways. Identification of specific EGFR mutation is critical in predicting response to TKI and successful lung cancer therapy. In both in vitro and in vivo studies, a combination of the ER antagonist and the EGFR tyrosine kinase inhibitor has been shown to decrease cell proliferation and tumor growth more than monotherapy either with ER antagonists or EGFR TKI [34,60,67]. Phase I clinical trial by Traynor and colleagues evaluated the efficacy of the gefitinib–fulvestrant combination. The results show improved overall survival in NSCLC patients with a mean of 65.5 weeks in those with higher ERβ tumor expression [143]. In female patients with stage IIIB/V NSCLC, the gefitinib–fulvestrant combination was ascertained as safe with considerable anti-tumor activity. The presence of a high percentage of cells within the tumors subjected to nuclear ERβ immunostaining was correlated with improved patient survival [144]. Additional studies have shown that combination therapy may increase the response duration in patients whose tumors harbor an EGFR mutation as well as an improved response rate in patients whose tumors are EGFR wild type. Studies on in vitro anti-estrogen administration revealed suppressed cell proliferation, stimulated apoptosis, suppressed tumor growth along with enhanced gefitinib sensitivity [67,149]. In addition, combined therapy (erlotinib + fulvestrant) was well-tolerated with greater clinical benefits than TKI monotherapy [148,184]. Recently, approximately ten clinical trials have evaluated the effects of anti-estrogen therapy alone or in the combination mode, mainly with TKI, and revealed significant clinical benefits in NSCLC patients. The results of these studies present significance for designing novel lung cancer treatment mechanisms (Table 2) [132].

## 10. Estrogen-Related Receptors and Non-Small Cell Lung Cancer

Estrogen-related receptors (ERRs) are a group of nuclear receptors including ERs. These receptors are implicated in breast cancer complications [62] and tamoxifen resistance in ER-positive breast cancer [63]. For the genetic characterization of ERRs, a repeated genetic analysis was undertaken. Based on the results, ERRs are grouped into the nuclear receptor 3B family (NR3B), comprising ERs, PRs, androgens, mineralocorticoids and glucocorticoids [185]. Three distinct ERRs have been identified to date. The genes responsible for ERR synthesis are ESRRA (*NR3B1*, ERRα), ESRRB (*NR3B2*, ERRβ) and G (*NR3B3*, ERRγ) [186]. Although ERRs exhibit a considerable structural and functional similarity with ERs, they do not have the capacity to bind with natural estrogens and are thus referred to as orphan nuclear receptors [64].

ERRα has been detected in the adult lung tissues both in normal physiology [64] and in different NSCLC stages. The corresponding expressions were noted as higher in NSCLC stages III and IV compared to those of I and II. The difference was estimated as statistically significant, irrespective of age, gender, pathological grade, T stage and distant metastasis [66]. No study has, to date, reported an ERRβ presence in the embryonic or adult human lungs. ERRγ expression has been reported in embryonic lung tissues including those of humans. However, it has not been detected in adult lungs [187]. ERRα has been implicated both in the proliferation and migration of NSCLC cells. Elevated ERRα expression has been noticed not only in NSCLC cells but also in xenograft NSCLC mouse models and clinical NSCLC samples. These results indicate a possible diagnostic or post-therapeutic prognostic role of ERRα in NSCLC [188,189,190]. The effects of ERRα knockout were examined on lung adenocarcinoma (LUAD) and lung squamous-cell carcinoma (LSCC) cells. The ERRα knockdown has resulted in LUAD cell synchronization at the G2-M phase transition, but the LSCC cells continued with cell-cycle progression [66]. These observations infer that while LSCC cell proliferation and growth are not affected by ERRα knockout, the LUAD cells were unable to propagate the cell cycle due to inhibited G2-M transition. These results indicate a cell-line-specific activity of ERRα [66]. The in vitro cell culture based experiments documented an ERRα role in cell proliferation and migration. An investigation by Makowiecki and colleagues monitored the effects of ERRα-specific inverse agonists/small interfering (si) RNA/shRNA on cell-cycle regulation [191]. In a similar observation, Wang and associates reported that in the in vitro cultured A549 NSCLC cells, ERRα affects the mitochondrial physiology and quenches ROS generation which suppressed the tumor suppressor protein p53 and pRB expressions. This caused an unhindered cell-cycle progression. The ERRα modulates multiple cell signaling pathways related to cell-cycle progression, together resulting in accelerated cell division and aggressive tumor progression [189]. A number of studies highlighted the ERRα involvement in aggressive epithelial–mesenchymal transition (EMT), together contributing to enhanced NSCLC cell migration. In one such attempt, Huang and colleagues [192] treated A549 NSCLC cells with ERRα inverse agonist XCT-790 and observed the ERRα involvement in epithelial–mesenchymal transition (EMT). Another significant investigation by Zhang and colleagues observed ERRα-induced pro-inflammatory cytokine IL-6 expression via NF-κB activation and cytoplasm to nucleus translocation [193]. Notably, it was previously demonstrated that di (2-ethylhexyl) phthalate (DEHP)-induced NSCLC migration and invasion needed an IL-6 upregulation [194,195]. These studies collectively indicated multiple important roles of ERRα in lung cancer proliferation and migration. Several synthetic antagonists of ERRs have been reported [196,197,198]. Additionally, numerous dietary-product-based ERRα agonists have been reported, namely cholesterol, flavones, daidzein, rutacarpine, apigenin and genistein [199,200,201]. The present era of individualized medicine mandates a screening of anti-ERR molecules alone or in combination with aromatase inhibitors (e.g., anastrozole), SERMs (e.g., tamoxifen) or SERDs (e.g., fulvestrant) in a specific NSCLC type. A comprehensive review article by *Mukherjee and collaborators* described various aspects of ERRα in NSCLCs pathophysiology [65].

## 11. Progress of Clinical Trials on Estrogen-Targeting-Driven Non-Small Cell Lung Cancer Treatment

Table 2 enlists the to-date progress of clinical trials aimed at the NSCLC treatment through targeting the estrogen-regulated signaling pathways and regulatory activities. Three of the described attempts have used a combinatorial approach, employing EGFR inhibitors, gefitinib or erlotinib, in combination with the estrogen-inhibiting compound fulvestrant. Since these studies correspond to different phases, the results do not present an unbiased picture to compare. For instance, a phase I clinical trial of gefitinib with fulvestrant exhibited a good safety profile with significant anti-tumor response in female sufferers having recurrent-stage NSCLC, irrespective of the therapies received. Three of the examined twenty patients in this study developed partial responses with a 15% response rate, 12-week median progression-free survival and a 38.5-week median overall survival. A striking observation corresponded to a longer overall survival in patients expressing ERβ to a higher extent, 65.5 weeks against the 21 weeks for low-ERβ-expressing tumors [143]. This was the first clinical trial of estrogen in NSCLC, the results of which were published in 2009, demonstrating safety and potential efficacy. Another of these studies was conducted at a phase II level, which found the clinical benefit rate (comprising partial responders and those with stable disease) to be higher amongst the sufferers treated with erlotinib and fulvestrant rather than erlotinib alone. This trend was specifically observed for the patients having EGFR wild configured tumors [144], albeit the third study (a phase II clinical trial) using this combination also observed a better response for the combinatorial mode but was restricted to NSCLC sufferers with EGFR wild configurations [184].

A far-sighted concluding generalization was made in the first of these attempts, wherein other than the feasibility of fulvestrant and erlotinib combined, no reliability was anticipated without knowing about the EGFR status. Thereby, administering erlotinib and fulvestrant in combination is indeed not suitable with respect to an unselected sufferer population [202]. Two studies focused on tuning hormonal response (estrogens via exemestane) in combination with immune checkpoint inhibitors (ICIs) and chemotherapeutic drugs (pemetrexed disodium and carboplatin), on both occasions in postmenopausal women [145,146]. A third configuration comprised of co-delivering EGFR and aromatase inhibitors, namely, gefitinib and anastrozole, taking encouragement from their synergistic response toward suppressed proliferation and apoptotic induction in NSCLC cell lines [149]. Amongst the remaining four attempts, three worked on optimizing the dosage extent of chemotherapeutic drugs (paclitaxel poliglumex with carboplatin, xentuzumab with abemaciclib and nintedanib with ethinyl estradiol and levonorgestrel) with or without hormonal therapies. The paclitaxel poliglumex co-delivery with carboplatin was a phase III study screened in >25 pg·mL^−1^ estradiol possessing female NSCLC sufferers [146] while the other two trials were phase I studies conducted irrespective of tumor stages with the xentuzumab–abemaciclib combination in the optional togetherness of hormonal replacement therapy (in progress) [145,147]. The remaining study analyzed the efficacy of entinostat pharmacokinetics with respect to specific food intake. This attempt co-delivered entinostat with erlotinib (an EGFR inhibitor) and exemestane (an anti-estrogen) to screen the pharmacokinetic distribution of entinostat, but no results have been published so far for this study. Of note, entinostat is a benzamide histone deacetylase inhibitor and is being screened for anticancer efficacy via multiple clinical trials. Thus, despite initial high hopes and preliminary success for the estrogen-targeting-mediated NSCLC treatment, challenges are high to accomplish reliable optimizations with respect to co-delivery attempts. Indeed, the rigors of phase I and II analysis with an accurate matching of in vitro studies could enhance the accuracy and minimize the fluctuating balance from in vitro to in vivo conditions.

## 12. Conclusions and Future Perspectives

Estrogens and associated receptors such as ERs, GPERs, EGFRs, orphan nuclear receptors and ERRs are involved in NSCLC development and progression. Besides endogenous estrogens, exogenous estrogens or estrogen similar compounds, such as phytoestrogens and xenoestrogens, may affect the aromatase expression and thereby estrogen synthesis as well as the expression intensity of multiple-estrogen-associated receptors. Thus, estrogen synthesis and modulation manifest as a high-risk factor for NSCLC development and complications. For estrogen-dependent growth-promoting signaling pathways, not only ERs (ERα and ERβ) but also GPER, EGFR and their downstream cell signaling pathways play multiple roles of significance. Additionally, the ERRs, orphan nuclear receptors of the 3B family in which ERs are also grouped, having close structural and functional similarities with ERs, are responsible for NSCLC complication. Thus, targeting estrogens and associated receptors either alone or in combination with other chemotherapeutic drugs may be a novel strategy for NSCLC treatment. The studies on in vitro cell culture experiments, xenograft mouse model along with clinical attempts, have collectively examined the efficacy of anti-estrogen or estrogen-associated receptor molecules against the stage-specific complicated NSCLCs. Collectively preliminary results of these experiments indicate beneficial effects of targeting estrogens, associated receptors and downstream cell signaling pathways against NSCLCs. Further research with careful thought-through concepts and advanced methods on environmental estrogens could enhance as well as improve the understanding for ascertaining the environmental influences. The interfaces of estrogen synthesis and ER expression along with their associated modulations of related signaling pathways can still provide a reliable and novel mechanism for a therapeutic confrontation of human cancers. Additionally, more carefully designed clinical studies with a large number of patients, correctly classified for ERs, associated receptor expression and their specific role in NSCLC; need to be evaluated in favor of treatment and survival response of NSCLC patients.

## Figures and Tables

**Figure 1 cancers-14-00080-f001:**
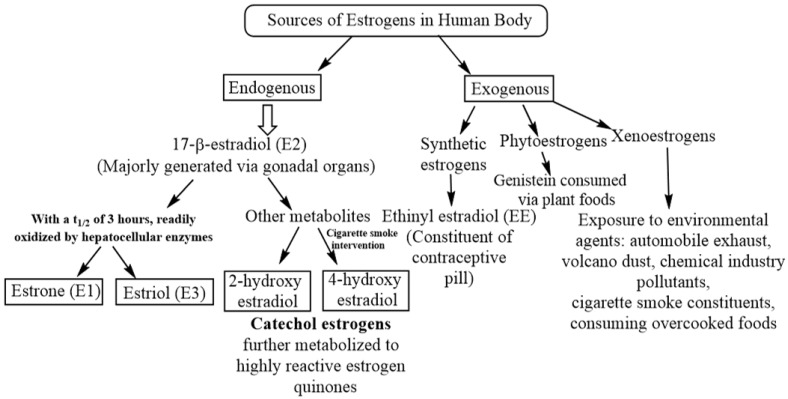
The different availability routes of estrogen in human body, distinguished into endogenous (within the body) and exogenous (outside the body). While exogenous sources present a greater diversity, endogenous estrogen is enabled through 17-β-estradiol (E2) and its metabolites.

**Figure 2 cancers-14-00080-f002:**
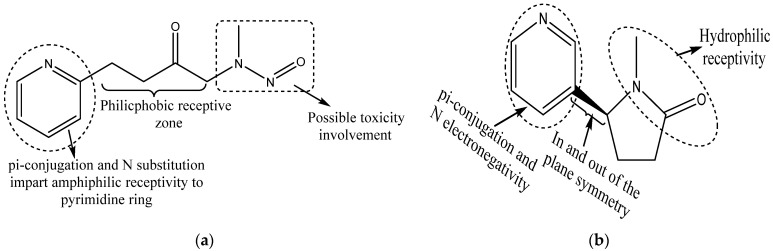
Chemical structures of (**a**) methylnitrosamino-pyridyl-butanone, a powerful carcinogenic agent involved in ERβ activation, and (**b**) cotinine, a potential nicotine metabolite that inhibits aromatase activity and exhibits sex-dependent activities. Presence of pi-conjugation with N substitution and double-bonded oxygen in close proximity with nitrogen are the indicative features of possible toxic responses.

**Figure 3 cancers-14-00080-f003:**
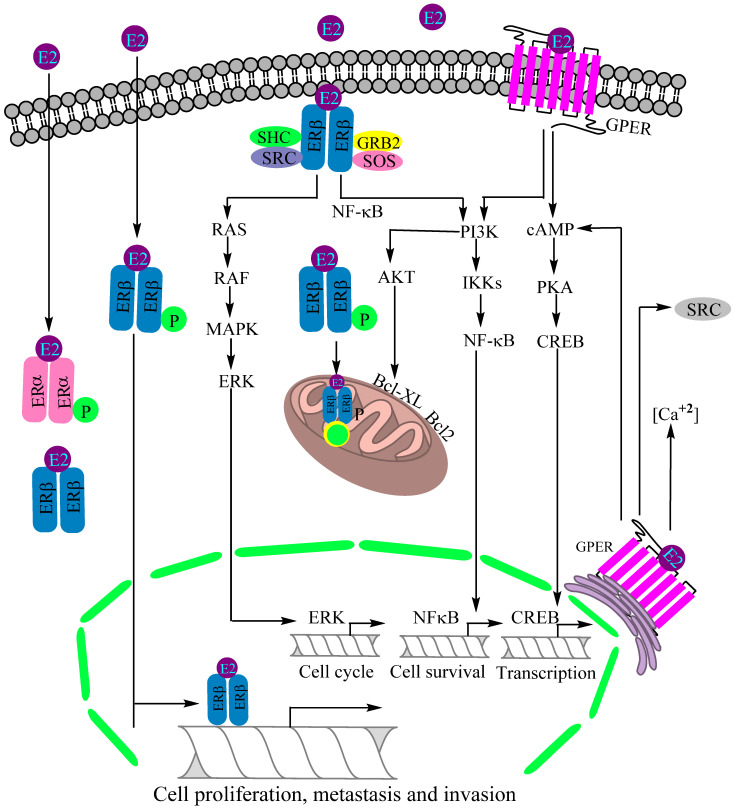
Recognized mechanisms of estrogen receptor (ER)-mediated lung cancer progression. Estrogen receptor β (ERβ) is the major estrogen receptor expressed in lung cancer, prevailing in cytoplasm, nucleus and mitochondria. The PI3K/IKK/NFκB, PI3K/AKT/Bcl-XL and RAS/RAF/MAPK/ERK are the activated signaling pathways for regulatory control of cell proliferation, invasion, metastasis, mitochondrial biogenesis and apoptotic suppression. The G-protein estrogen receptor (GPER) is activated by estradiol (E2) binding to intercept the cAMP/PKA/CREB and PI3K/IKK/NFκB signaling pathways, together modulating a neoplastic transformation. Abbreviations: PKA: Protein kinase A, CREB: cAMP response-element-binding protein, IKK: Inhibitor of nuclear factor-κB (IκB) kinase, MAPK: Mitogen-activated protein kinase, ERK: Extracellular signal-regulated kinase, AKT: Protein kinase B, SRC: Steroid receptor coactivator, SHC: SHC-transforming protein 1, GRB2: Growth factor receptor-bound protein 2, SOS: DNA repair system.

**Figure 4 cancers-14-00080-f004:**
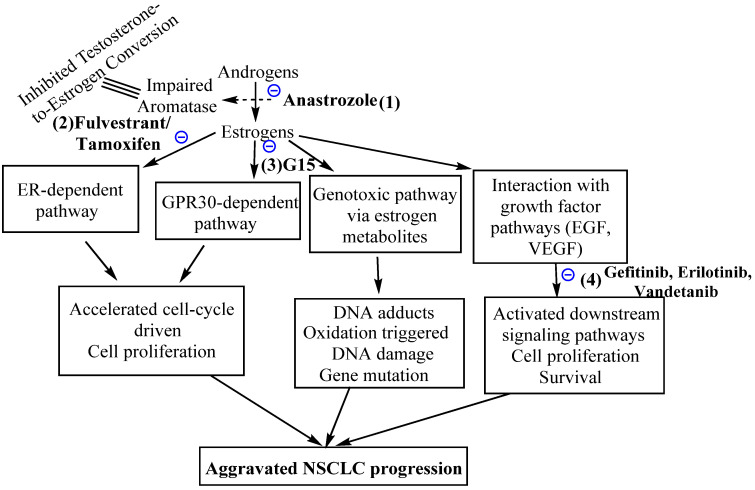
Possible NSCLC treatment mechanisms for estrogen signaling pathway targeted NSCLC treatment, comprising (1) aromatase suppression using the inhibitor anastrozole, (2) impaired activity of ERs using pure anti-estrogen fulvestrant, (3) impaired GPR30-driven signaling via antagonist G15 and (4) targeting the estrogen-activated growth factor pathways, mainly EGF and VEGF, using gefitinib, erlotinib and vandetanib drugs. These strategies can be used in singular as well as in combination mode.

**Figure 5 cancers-14-00080-f005:**
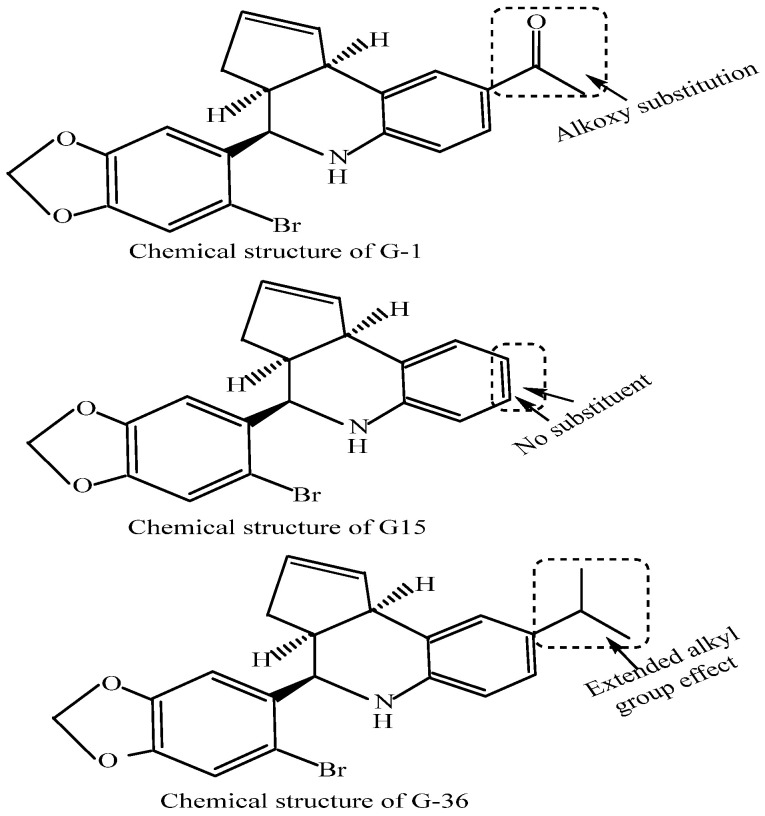
Chemical structures of the GPER antagonists, G-1, G-15 and G-36. The structures of these compounds distinguish an influence of extended alkyl, alkoxy and null substitution in the terminal rings. Though G-36 has been reported only recently for its reduced non-specific response, the G-15 has been listed in 2019 only for its anti-estrogen-mediated control of NSCLC growth.

**Figure 6 cancers-14-00080-f006:**
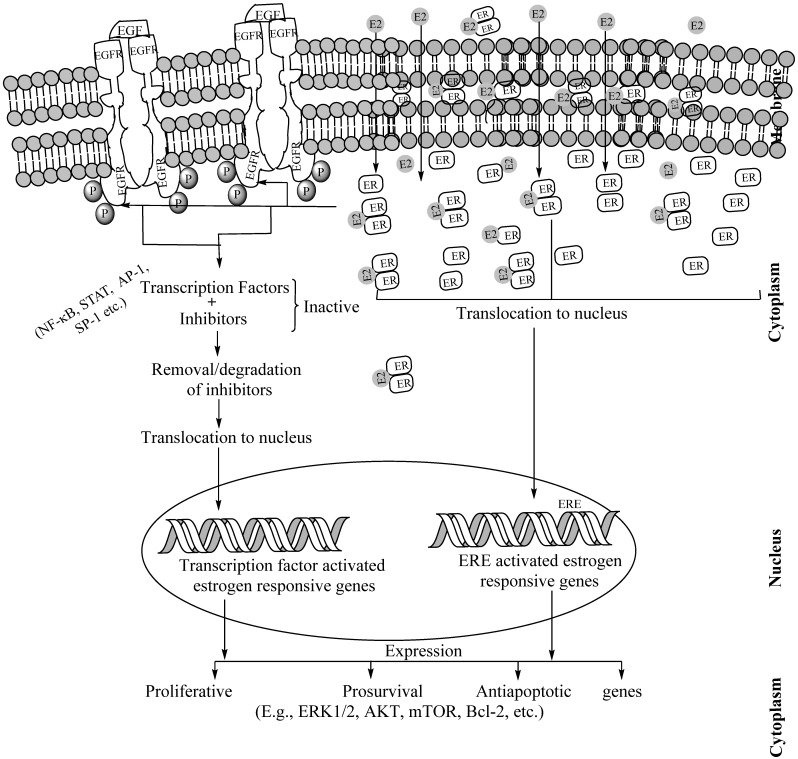
Schematic description of EGFR–ER interactions, initiated by ER dimmers on binding the carboxyl terminal region of EGFR in the cytosol. While EGFRs are transmembrane proteins, ERs are present in the membrane caveolae, cytoplasm including endoplasmic reticulum and mitochondria as well as in the nucleus. The carboxy terminus domain of cytoplasmic EGFR directly communicates with E2-ER dimmers. This cross-talk leads to the activation of estrogen-responsive genes in the following two ways. In the first step, the E2-ER dimer translocates to the nucleus and binds to the estrogen-response elements (EREs) of the estrogen-responsive genes and activates them for transcription. In the subsequent step (in absence of ERE), the estrogen-responsive genes, the EGFR-E2-ER cross-talk activates various transcription factors, translocates them to the nucleus, binds to the specific transcription factor binding sites of the estrogen-responsive genes and activates them for transcription. This transcriptional activation through ERE-dependent or -independent manner leads to the expression of pro-proliferative, prosurvival and antiapoptotic proteins which complicates the tumorigenesis. Of note, E2-ER dimmers and EGF–EGFR interaction have functions independent of their cross-talk. Abbreviations: ER: Estrogen receptor. ERE: Estrogen Response Elements, E2: Estradiol, ECM: Extracellular matrix.

**Table 1 cancers-14-00080-t001:** The different sources of estrogens, distinguished via endogenous and exogenous origins.

Type of Estrogen	Sources
Natural (Endogenous) (Estrone, 17β-estradiol, estriol, estretrol)	Extensively by gonadal organs (ovary/testis), low-level production in various organs due to aromatase generation (e.g., lungs, brain, etc.)
Synthetic (Exogenous) (Ethinyl estradiol, diethylstilbestrol, estradiol valerate, estropipate, conjugate esterified estrogen and quinestrol)	Either as constituent of contraceptive pills or hormone response therapy
Phytoestrogen (Exogenous) (Genistein, coumestans, quercetin, isoflavones, flavones, lignans, saponins and stilbenes)	Plant foods such as soy beans, tofu, tempeh, soy beverages, linseed (flax), sesame seeds, wheat (as lignans), berries (resveratrol), oats, barley, dried beans, lentils, rice, mung beans, apples, carrots, wheat germ, ricebran, soy linseed bread.
Xenoestrogen (Exogenous) (Bisphenol A (BPA), Dichlorodiphenyltrichloroethane (DDT), polychlorinated biphenyl (PCB), heavy metals, phthalates, alkylphenols, epileptic drugs	Plastics (water bottles, disposable cups, plastic wrap, food containers), pesticides (used on non-organic fruits and vegetables), tap water (chlorine and runoff byproducts), chemicals in cosmetics, lotions, shampoos and other body care materials

**Table 2 cancers-14-00080-t002:** Current progress of clinical trials aimed toward using estrogen-expression-targeted non-small cell lung cancer treatment.

Clinical Trial Registry	Primary Objective of the Trial	Phase of Study, Tumor Stage and Current Status	Findings Published (Ref.)
NCT01556191	Evaluating an EGFR tyrosine kinase inhibitor (EGFR-TKI), gefitinib and an EGFR-TKI-anti-oestrogen (erlotinib, fulvestrant) combined potency in women with advanced-stage non-squamous lung cancer	Phase I, stage IV lung cancer, completed	Improved outcome [143]
NCT00100854	Evaluation of synergistic fulvestrant delivery with erlotinib for the non-small cell lung cancer (NSCLC) treatment	Phase II, stage IIIB or IV non-small cell lung cancer, completed	Improved outcome [144]
NCT02666105	Evaluation of adding exemestane therapy in postmenopausal women suffering from NSCLC while on treatment with an immune checkpoint antibody (pembrolizumab, atezolizumab or nivolumab)	Phase II, advanced stage NSCLC, ongoing	Improved outcome [145]
NCT01664754	Determining the safety and tolerability of escalating exemestane doses on being co-delivered with pemetrexed (pemetrexed disodium) and carboplatin in postmenopausal womensuffering from NSCLC	Phase I, stage IV non-squamous NSCLC, ongoing	Combination is safe and well-tolerated, response rate correlates with tumor aromatase expression [146]
NCT02751385	Screening the effect of nintedanib on the (ethinylestradiol + levonorgestrel) pharmacokinetics in NSCLC patients	Phase I, all NSCLC patients, completed	No findings published todate
NCT00576225	Screening the effect of paclitaxel poliglumex (CT-103)/carboplatin versus paclitaxel/carboplatin for women NSCLC sufferers	Phase III, sufferers having >25 pg·mL^−1^ estradiol, completed	CT-103 did not provide superior survival over the paclitaxel-carboplatin for first-line treatment of NSCLC patients, results were comparable for progression-free and overall survival [147]
NCT03099174	Ascertaining a safe dosage of xentuzumab in combination with abemaciclib with or without hormonal therapies in lung and breast cancer	Phase I, no stage distinction, ongoing	Findings not yet published
NCT00592007	Screening the impact of adding fulvestrant to erlotinib in NSCLC patients	Phase II, stage IIIB or IV, concluded	[148]
NCT00932152	Fulvestrant and anastrozole (aromatase inhibitor) as consolidation therapy in postmenopausal women NSCLC sufferers	Phase II, advanced stage NSCLC, concluded	[149]
NCT01594398	Assessing the food effect on entinostat pharmacokinetics in NSCLC sufferers (ENCORE110)	Phase I, no stage distinction, completed	No findings published, study listed from [141]
AM2013-4664	Evaluation of erlotinib antitumor activity in NSCLC on fulvestrant inclusion in the patients received > 1 chemotherapy regimen	Phase II, advanced state NSCLC patients	[150]

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
