# Peer review of "Targeting Estrogens and Various Estrogen-Related Receptors against Non-Small Cell Lung Cancers: A Perspective"

_cancers, 2021, doi:10.3390/cancers14010080_

Round 1

Reviewer 1 Report

This review is an interesting work, well structured, and comprehensive. It covers all aspects of estrogen action in NSCLC. I do not have major concerns and I support its publication.

I suggest a few minor corrections for the authors:

  • NSCLC is often described as "complicated state" or "complications. Please define if you refer to the advanced stage or whatever.
  • On page 8, at the end of paragraph 5, you should mention Table 2 when you discuss clinical trials. Similarly, on page 13 paragraph 9.
  • on page 10, after reference 103, the phrase "various human tissues including the lungs are the subjects to E2-GPER actions" should be rephrased (maybe "subjected to").
  • On Page 11, It has been written "(To write one line here)". I do not understand the meaning of this phrase. 

Author Response

The manuscript has been adjudged on the following criteria, each awarded a scoring on 5 point scale.

Is the work a significant contribution to the field?         Score:  4 

Is the work well organized and comprehensively described?              Score:  5 

Is the work scientifically sound and not misleading?    Score:  4 

Are there appropriate and adequate references to related and previous work? Score:  5     

Is the English used correct and readable?   Score:  4 

GENERAL COMMENTS AND SUGGESTIONS FOR AUTHORS FROM REVIEWER 1

This review is an interesting work, well structured, and comprehensive. It covers all aspects of estrogen action in NSCLC. I do not have major concerns and I support its publication.

OUR RESPONSE: Thanks for reviewing our manuscript

SPECIFIC COMMENTS OF REVIEWER 1:                                                   

I suggest a few minor corrections for the authors:

Comment No. 1: NSCLC is often described as "complicated state" or "complications. Please define if you refer to the advanced stage or whatever.

Our response: In general, by complication, we meant severity or aggressiveness of diseased state. Three studies quoting complicated state of NSCLC (references 49, 66 and 122) are now modified after matching the stages of NSCLC studied. For references 49 and 122 (Section 4, Page number 9), the investigation was made on cultured NSCLC cells while for reference 66, the investigation analyzed comparative ERα expressions in stages, I, II, III and IV. Now the sentence on Page Number 18 in the revised manuscript (Section 10), is accordingly modified. 

Comment No. 2: On page 8, at the end of paragraph 5, you should mention Table 2 when you discuss clinical trials. Similarly, on page 13 paragraph 9.

Our response: Now the mention of Table 2 is adequately made at the end of section 5 (on Page number 10 in the revised manuscript) and towards the end of section 9 (Page number 18 in the revised manuscript).

Comment No. 3: On page 10, after reference 103, the phrase "various human tissues including the lungs are the subjects to E2-GPER actions" should be rephrased (maybe "subjected to").

Our response: We are sorry for this grammatical lapse. Now the words, “subjects to” (appearing originally on Page Number 10) have been rephrased as “subjected to” (on page number 13, second paragraph from top, in the revised manuscript), in the line next to the one with reference 103.

Comment No. 4: On Page 11, It has been written "(To write one line here)". I do not understand the meaning of this phrase. 

Our response: We are sorry for this lapse, Sir. Now the referred line (originally on Page Number 11) has been replaced by a newly phrased sentence. This sentence appears on Page number 14 in the revised manuscript. The sentence is as below:

The mechanisms relates to the inhibition of estrogen action by the competitive inhibitors of estrogen receptors (e.g tamoxifen), degradation of estrogen receptors (e.g fulvestrant or ICI182780) or inhibition of estrogens synthesis by aromatase inhibitors (e.g. anastrazole). Other targets of inhibition are GPER (e.g. by G15, G36) or EGFR by EGFR inhibitors (e.g gefitinib, erlotinib).

Reviewer 2 Report

In the present review, the role of the estrogen receptors in NSCLC progression is explored, with a look also at some clinical studies. First, the manuscript needs a careful revision as in its present form it is poorly written. The paper appears to be a list of unrelated information (sometimes without mechanistic explanations), which in some cases are not relevant or whatever not appropriate to the main aim of the review. For instance, why the authors largely focused on the GPER agonists and antagonists if only one paper on G15 in NSCLC is discussed? Similarly, the thread of the clinical trials discussed is not clear. On the other hand, the efficacy of anti-estrogens molecules against NSCLC should be further explored.

Author Response

REVIEWER 2

EVALUATION OF REVIEWER 2 REGARDING THE STANDARD OF THE MANUSCRIPT:                                                     

Is the work a significant contribution to the field?         Score:  2 

Is the work well organized and comprehensively described?              Score:  1 

Is the work scientifically sound and not misleading?    Score:  3 

Are there appropriate and adequate references to related and previous work? Score:  4     

Is the English used correct and readable?   Score:  2 

GENERAL COMMENTS AND SUGGESTIONS FOR AUTHORS FROM REVIEWER 2

In the present review, the role of the estrogen receptors in NSCLC progression is explored, with a look also at some clinical studies. First, the manuscript needs a careful revision as in its present form it is poorly written. The paper appears to be a list of unrelated information (sometimes without mechanistic explanations), which in some cases are not relevant or whatever not appropriate to the main aim of the review. For instance, why the authors largely focused on the GPER agonists and antagonists if only one paper on G15 in NSCLC is discussed? Similarly, the thread of the clinical trials discussed is not clear. On the other hand, the efficacy of anti-estrogens molecules against NSCLC should be further explored.

Our response:

Thanks reviewer 2 for his/her critical comments. The comments are very general regarding the overall standard of the manuscript. There are no specific comments from reviewer 2. However, reviewer mentioned some of the general shortcomings of this manuscript.

Here is the response of those comments:

Reviewer 2 Comment: First, the manuscript needs a careful revision as in its present form it is poorly written.

Our response: On receiving the invitation from the Executive Editor of this special issue for contributing a manuscript we have communicated him through email messages. As we understand from the communication with the Executive Editor of this special issue is that the specific goal of this special issue is to focus on the effects of environmental estrogens including phytoestrogens as well as estrogens produced by the mammalian body on the complication of various types of cancers. Regarding estrogens dependent cancer complications estrogen receptors takes a very important role. In fact, anti-estrogen molecules are one of the most important targets of estrogen dependent cancer therapy. Based on our knowledge we have very specifically described the role of estrogens (environmental, internally produced) on NSCLC complications. In this aspect the role of various estrogens associated receptors including, ERs, EGFR, GPERs and orphan nuclear receptors ERR are discussed. Thus, we believe very strongly that in the current form it is impossible for us to revise the manuscript without changing the focus of this specific topic.

Reviewer 2 Comment: The paper appears to be a list of unrelated information (sometimes without mechanistic explanations), which in some cases are not relevant or whatever not appropriate to the main aim of the review. For instance, why the authors largely focused on the GPER agonists and antagonists if only one paper on G15 in NSCLC is discussed?

Our response: It is now widely accepted that GPERs activities are influenced by estrogens  including environmental estrogens and phytoestrogens. GPERs are also cross-talks with estrogen receptors. Both in normal lungs as well as under NSCLC complication GPERs are expressed in the lung tissues. Thus, GPERs are natural choice of discussion of this review article. However, it is true that at present few studies are available regarding the effects of GPER inhibitors against NSCLC complications. So, in near future more studies in this particular topic may enhance our knowledge in this specific subject.

Reviewer 2 Comment: -----Similarly, the thread of the clinical trials discussed is not clear.

Our response:  In the present manuscript 10 clinical trials are presented in a tabulated form which discussed regarding targeting of estrogens and/ or estrogen associated receptor molecules against NSCLC.  In the table clinical trial registry, primary objective of the trial, phase of study, tumor stage and current status as well as references are presented. Four of the trials mentioned improved outcome, one of them mentioned that the trial was safe, one of them mentioned that the trial did not improve the outcome and rest of the four studies are yet to publish their data. To the best of our knowledge the tabulated presentations of the clinical trials suitably fit with the present manuscript.

Reviewer 2 Comment- - - - On the other hand, the efficacy of anti-estrogens molecules against NSCLC should be further explored.

Our response: In the last couple of decades, thousands of papers have published the effects of anti-estrogens or anti-estrogen receptor molecules towards reduced complications of various diseases including lung cancers.

On page number 9 (in the revised manuscript), the section “4. Role of estrogen receptors in non-small cell lung cancer complication” (on Page number 6 in the original manuscript) described the role of estrogen receptors on NSCLC complications.

On page number 8 (in the original version of manuscript) of this publication entitled “5. Efficacy of anti-estrogen and anti-estrogen receptor molecules against non-small cell lung cancers” (Page number 10, in the revised manuscript) described the efficacy of anti-ER molecules against NSCLC complication.

On page number 13 (in the original version) of this manuscript entitled, “8. Interactions of estrogen receptors and epidermal growth factor receptors for aggravated tumorigenesis in non-small cell lung cancers” (on Page Number 16 in the revised manuscript) described the effects of interactions of ER and EGFR on NSCLC complications.

Additionally, on page number 13 (in the original manuscript) of this manuscript entitled, “9. Potentials of dual targeting of estrogen receptors and epidermal growth factor receptors against non-small cell lung cancers” (on Page number 16 in the revised manuscript) the effects of dual targeting of ER and EGFR in reducing the complications of NSCLC has been discussed.

Since the current length of the manuscript appropriately discussed the role of ERs and anti-ER molecules against the complications of NSCLC and the manuscript is not solely dedicated on ERs or anti-ER molecules against NSCLC we believe further enlarging of the manuscript in this particular area may be unnecessary.

Reviewer 3 Report

The review was aimed to describe the role of the estrogens and their receptors in lung cancer pathogenesis and therapy. The manuscript also presents the current and prospective view of NSLC therapies targeting ER-mediated pathways. 

In general, the manuscript is well-written. The authors consequently describe the types and the sources of the estrogens, the physiological role of estrogens receptors (ERs) in lung function, and non-small lung cancer (NSLC) pathogenesis. The efficiencies and perspectives of anti-estrogen inhibitors and ERs antagonists are also described.   

I suggest including more details in some aspects of estrogen's role in lung carcinogenesis. 

For example, the authors mention that estrogen metabolites exhibit mutagenic and carcinogenic effects and refer to paper 60. This should be described in more detail.

Similarly, the ability of estrogen's metabolites to convert protooncogenes to the oncogenes should be cleared and described more precisely. 

The phrase "endocrine disruptors" is also not suitable for the scientific manuscript and is required to be explained in detail to illustrate what kind of pathways are being dysregulated.   

Chapter 8 illustrating the potential cross-talk between ER and EGFR is intriguing but composed of multiple reports illustrating the opposite data. Therefore, the best way will be to make another figure illustrating the cross-talk between these pathways and thereby providing the molecular basis for perspective targeted therapies based on dual targeting of ERs and EGFR for NSLC ( as was shown in Chapter 9)

Author Response

Reviewer 3:                                                      

Is the work a significant contribution to the field?         Score:  5 

Is the work well organized and comprehensively described?              Score:  5 

Is the work scientifically sound and not misleading?    Score:  4 

Are there appropriate and adequate references to related and previous work? Score:  4     

Is the English used correct and readable?   Score: 4 

Comments and Suggestions for Authors

The review was aimed to describe the role of the estrogens and their receptors in lung cancer pathogenesis and therapy. The manuscript also presents the current and prospective view of NSLC therapies targeting ER-mediated pathways.

In general, the manuscript is well-written. The authors consequently describe the types and the sources of the estrogens, the physiological role of estrogens receptors (ERs) in lung function, and non-small lung cancer (NSLC) pathogenesis. The efficiencies and perspectives of anti-estrogen inhibitors and ERs antagonists are also described.  

I suggest including more details in some aspects of estrogen's role in lung carcinogenesis.

Comment No. 1: For example, the authors mention that estrogen metabolites exhibit mutagenic and carcinogenic effects and refer to paper 60. This should be described in more detail.

Our response: Now, the mutagenic and carcinogenic effects of estrogen metabolites are described in the Introduction (Page number 3 in the revised mansucript), after the mention of references 3 and 34.  The added text is as below:

After reference 3:

The common links between estrogenic metabolism and tobacco combustion aggravate the carcinogenic actions in the lungs. The constituents of cigarette smoke activate cytochrome P450 1B1 (CYP1B1), the enzyme mitigated in estrogen metabolism alongwith the synthesis of corresponding catecholic derivatives. Intermediates and products formed therein accumulate as reactive oxygen species (ROS), concurrently prevailing as DNA adduct that collectively tampers the genetic material. Interactions of estrogens with cigarette smoke constituents is driven via ER interception, which forms genotoxic metabolites including 4-hydroxyestrogen (4-OH-E2), 4-hydroxyestrone (4-OH-E1) or estrogen’s quinine derivatives. The process is viciously regulated by the CYP1B1 activity that controls E2 metabolism and concurrent interaction of products thereof with cigarette smoke carcinogens which on further transformation, aggravate the ROS formation. Catalyzing the hydroxylation at 2 and 4 positions of E1 and E2, the generated 4-hydroxylated metabolites are carcinogenic [22, 23]. Erstwhile of being formed at once, the endogenous catechol estrogens can be oxidized by any enzyme having an oxidative ability, generating the vulnerable electrophilic estrogen o-quinones and semiquinones. These quinine derivatives aggravate the ROS formation through a series of redox cycling events and are detrimental to cells in multiple manners. For instance, O-quinone metabolism indirectly generates free hydroxyl radicals, the most harmful oxidizing agents. These molecules induce DNA damage via multiple mechanisms, such as inducing single strand breaks, chromosomal aberrations and 8-oxo-2’-deoxyguanosine formation. The quinines and semiquinones can also inflict a direct cellular DNA damage via forming adducts, culminating in genotoxic effects (depurination). Studies have highlighted the capability of catechol estrogens, o-quinones or their metabolites which bind to ER and on further transportation to ERE in the nucleus, result in DNA mutation via free radical emission [24, 25]. A prospect of further caution pertains to overexpressed CYP1B1 which is enacted through long lasting tobacco combustion. Thereby, the risk of concurrent free radical and ROS induced damage is higher in chain smokers or those with a long smoking association.

After reference 34:

Studies reveal a more than twice higher adenocarcinoma risk in the smoking females advised or undergoing estrogen replacement therapy (ERT) than those not subjected to ERT, odds proportion being 32.4 to 13.1. Contrary to this, non-smoking women taking ERT exhibited no significant adenocarcinoma risk [35]. Lately though, the impact of HRT for a possible lung cancer aggravation has revealed a 50% higher risk for the women under a combined hormone therapy (estrogen with progestin) [36-39]. Besides increased lung cancer risk, the HRT mediated via estradiol and progesterone combine also exhibited a significant association between both a younger median age for lung cancer diagnosis and shorter median survival duration [39-41]. Apart from CYP1B1, the CYP1A1 gene encodes for phase I enzyme which metabolizes polycyclic aromatic hydrocarbons (PAH) in tobacco smokes, preventing the precarcinogen from turning carcinogenic [42]. The circulating steroid hormones in females impair this CYP1A1 action due to their interaction with receptors in the lungs of sufferers. As a result, the carcinogen formation becomes unregulated which aggravates the LC risk. Studies in animals support above predictions with female mice being more sensitive to chemically induced lung carcinogenesis, which tend to be inhibited by ovariectomy [43]. In a study monitoring the implicit activities, Hammoud and colleagues examined the lung tumor intensity subsequent to estradiol intake (2 μg per day for 10 weeks). While tumor count and volume declined on ovariectomy contrary to intact female mice, estradiol administration increased the tumor count and volume in the ovariectomized mice compared to untreated female mice. Similar distinctions were observed in male mice administered the equal E2 concentration [44].   

Comment No. 2: Similarly, the ability of estrogen's metabolites to convert protooncogenes to the oncogenes should be cleared and described more precisely.

Our response: Though no substantial studies have till date, explained about the estrogen metabolite mediated protooncogene to oncogene conversion in non-small cell lung cancer (NSCLC) but still the molecular aspects of proto-oncogenes involved in aggravating NSCLC are described on Page numbers 5 and 6 (in the revised manuscript), in section 2 (originally on Page numbers 4 and 5): Exposure of mammalian lungs to endogenous and exogenous estrogens including synthetic estrogens, phytoestrogens and xenoestrogens (in the revised manuscript) highlighted in red. The added lines are as under:

Development of lung cancer in humans may be associated with gene deletions on the chromosomes 1, 3, 11, 13 and 17. Prominent protooncogenes induced by these genetic aberrations include c-jun, ras and c-raf1 alongwith a loss of tumor suppressor gene, p53. Amongst these, c-erbB-2, c-sis and c-fes are the prominently expressed or missing in the NSCLC and can be of substantial importance in the selection of differentiation pathway. Preclinical studies have revealed major NSCLC proliferative activities regulated through genomic and indirect non-genomic mechanisms of E2. The c-myc oncogene is often noted as being amplified in small-cell lung cancer (SCLC) cell lines. Investigations demonstrate the ER influence on NSCLC cells mediated via EGFR signaling driven cell cycle regulation, the cAMP, MAPK, AKT pathways and the promotion of c-myc and cyclin D expressions [79, 80]. Despite well-known for estrogen induction, the c-myc promoter does not possess any estrogen-response element (ERE).

While exact investigations tracing estrogen response in NSCLC are rather too scarce, the results from the modulated expressions in other tumors (primarily breast) indeed offer a basis of functional link. A 2011 study of this kind discusses the yester experience of distant elements being involved in estrogen induced gene expression. Analysis revealed insignificant effect on c-myc proximal promoter activity, though it stimulated the activity of a luciferase reporter characterized by a distal 67 kb enhancer. This activity of estrogen was noted as dependent on a half estrogen-response element apart from an activator protein (AP-1) site residing within this enhancer. The conservation of this AP-1 binding site in 11 distinct mammalian species and active estrogen-AP-1, jointly suggest AP-1 as the source to propagate the tumor development. Exclusive involvement of AP-1 in this tumor promoting activity was confirmed by the small interfering RNA experiments and chromatin immunoprecipitation assays wherein estrogen receptor-AP-1 cross talk was demonstrated as an essential factor to induce c-myc expression [81]. Noted human lung cancer cell lines are characterized by the generation of multiple growth factors that are engaged in proliferation via paracrine and autocrine loops through specific receptors. The resultant products from some activated protooncogenes (c-sis and c-erbB-2) are the homologous sequences to some specific growth factor (such as, platelet derived growth factor, PDGF) and the EGFR, commonly identified in lung cancer. Actions of these growth factors serve as decisive links for tumor progression via intracellular signal transduction and specific oncogenic activation. The observations of a 1990 study mention the presence of fur gene in 32 of the 40 NSCLC examined biopsies [82]. The fur gene encodes for a membrane associated protein and its involvement towards NSCLC development is due to location immediately upstream of the fes/fps protooncogene. With its receptor like attributes, the fur gene together with fes/fps proto-oncogene, exhibits a tyrosine protein kinase activity, which draws significance from the therapeutic rationale of using selective tyrosine kinase inhibitors of EGFR, whose overexpression remains a recognized hallmark of NSCLC [83]. The above 1990 investigation examined c-erbB-2 expression in 60 patients (SCLC and NSCLC both) alongwith 11 lung cancer cell lines. The product from c-erbB-2 is a membrane protein having tyrosine kinase activity with c-erbB-2 having a sequence homology with EGF receptor. The gene for c-erbB-2 was mapped to chromosome 17 (17q21) wherein only two of the 60 examined biopsies exhibited amplification. Interestingly, all the seven investigated NSCLC cell lines showed an increased c-erbB-2 gene expression with characteristically high transcription extent compared to the SCLC cell lines. With a sequence homology of EGF receptor, for which the estrogenic activity is well-known for aggravating tumorigenesis via EGFR signaling, the c-erbB-2 gene presents a high likelihood of being intercepted by estrogen for an enhanced carcinogenic response. A different study herewith, noticed enhanced c-erbB-2 protooncogene expression in squamous and large cell undifferentiated cancers contrary to the adenocarcinomas. Encoding a growth factor receptor on glandular epithelium, this gene is implicitly amplified only in an adenocarcinoma. Abnormal expression of this gene is noticed at a higher rate in advanced than the early stage cancers [84]. This investigation studied the expressions of c-myc, c-myb, c-ras-Ha, c-erbB-1, c-erb-B-2 protooncogenes in NSCLC and observed their detectable abnormalities in adenocacinomas (ten out of sixteen) and large cell cancers (two out of two).

The c-sis oncogene in the same study [83], was noticed as prevailing homologous to platelet derived growth factor (B-chain), expressed in five of the six examined non-SCLC cell lines. Transcripts for PDGF A-chain were detected in all studied six non-SCLC cell lines. The Transforming Growth Factors (TGFs) α and β were positive in four and five NSCLC cell lines. Injection of studied NSCLC cell lines into nude mice revealed varying extents of collagen rich tumor stromata, suggesting a paracrine functioning of PDGF and TGF in non-SCLC cell lines. Here again, five of studied six NSCLC cell lines possessed EGF receptor transcripts. Thereby, a significant association of c-erbB-2, c-sis and c-fes with EGFR expression infers an aggravated NSCLC carcinogenesis via ER-EGFR interactions. So, the NSCLC aggravation is the outcome of either the transcription factor activation followed by translocation to nucleus (activation) or via direct binding with ERE promoter sensitive locus of estrogen responsive genes (ERG).

Comment No. 3: The phrase "endocrine disruptors" is also not suitable for the scientific manuscript and is required to be explained in detail to illustrate what kind of pathways are being dysregulated. 

Our response: Now the meaning of ‘endocrine disruptors’ is defined on Page number 2 (in the Introduction) in the revised manuscript, just after the first mention of endocrine disruptive properties, line quoted with reference 17. The included explanation is also supported by three additional references. The added text is as below:

Endocrine disruptors are the molecules which regulate the expression of multiple genes, either via direct interaction with ERE or indirectly via interacting with transcription factors, including members of activator protein-1 (AP-1), nuclear factor-κB (NF-κB), signal transducer and activator of transcription (STATs) and families of specificity protein (SP-1) or by modifying the estrogen metabolism [18-20].

Comment No. 4: Chapter 8 illustrating the potential cross-talk between ER and EGFR is intriguing but composed of multiple reports illustrating the opposite data. Therefore, the best way will be to make another figure illustrating the cross-talk between these pathways and thereby providing the molecular basis for perspective targeted therapies based on dual targeting of ERs and EGFR for NSLC ( as was shown in Section 9).

Our response: Now, a figure (Fig.6) illustrating the ER-EGFR cross-talk, with respect to aggravated lung carcinogenesis is added. The Figure is inserted in Section 8 of the revised manuscript on Page Number 16. The figure demonstrates that estrogen receptors could aggravate the lung cancer pathogenesis, either through EGFR interaction driven activities on transcription factors (in the cytosol) or directly intercept the estrogen receptor genes (in the nucleus) which have an active site for estrogen receptor element (ERE) sequence in their promoter region.

Round 2

Reviewer 2 Report

NA